# Induction of CXCL10-Mediated Cell Migration by Different Types of Galectins

**DOI:** 10.3390/cells10020274

**Published:** 2021-01-30

**Authors:** Dina B. AbuSamra, Noorjahan Panjwani, Pablo Argüeso

**Affiliations:** 1Schepens Eye Research Institute of Massachusetts Eye and Ear, Department of Ophthalmology, Harvard Medical School, Boston, MA 02114, USA; pablo_argueso@meei.harvard.edu; 2New England Eye Center/Department of Ophthalmology, Tufts University Medical School, Boston, MA 02111, USA; Noorjahan.Panjwani@tufts.edu

**Keywords:** CXCL10, chemotaxis, monocytes, T cell, galectins

## Abstract

Chemokines are an extended group of chemoattractant cytokines responsible for the recruitment of leukocytes into tissues. Among them, interferon-γ-inducible protein 10 (CXCL10) is abundantly expressed following inflammatory stimuli and participates in the trafficking of monocytes and activated T cells into sites of injury. Here, we report that different members of the galectin family of carbohydrate-binding proteins promote the expression and synthesis of CXCL10 independently of interferon-γ. Interestingly, CXCL10 induction was observed when galectins came in contact with stromal fibroblasts isolated from human cornea but not other cell types such as epithelial, monocytic or endothelial cells. Induction of CXCL10 by the tandem repeat galectin-8 was primarily associated with the chemotactic migration of THP-1 monocytic cells, whereas the prototype galectin-1 promoted the CXCL10-dependent migration of Jurkat T cells. These results highlight the potential importance of the galectin signature in dictating the recruitment of specific leukocyte populations into precise tissue locations.

## 1. Introduction

Chemokine interferon-γ-inducible protein 10 (CXCL10) is a chemoattractant cytokine originally identified in the nineteen eighties by Luster et al. as a 10 kDa secreted polypeptide [1,2]. Its expression was observed in a number of immune and non-immune cells as an early and transient response to interferon-γ stimulation [2]. Subsequent studies have shown that, once secreted, CXCL10 binds to CXCR3, a seven transmembrane pass G protein-coupled receptor, on the surface of immune cells to exert a number of biological processes during the inflammatory response [3]. These include participation in leukocyte trafficking into the inflamed tissue as well as modulation of innate and adaptive immune responses. CXCL10 is known to contribute to the pathophysiology of several disease states. Increased CXCL10 levels have been observed in tissue specimens and serum of patients with autoimmune disorders [4], chronic inflammation [5], infectious disease [6] and cancer [7].

There is ample evidence supporting a role for galectins, a family of small soluble multivalent proteins previously termed S-type lectins, in the regulation of the inflammatory response. These proteins can stimulate the production of pro- and anti-inflammatory cytokines when bound to cells, and influence the localization of immune cells by creating chemotactic gradients [8,9]. A common feature of all galectins is the presence of conserved carbohydrate-recognition domains (CRDs) with affinity towards glycoconjugates containing β-galactose. According to structure, they have been classified into three major groups; (i) prototypical, with a single CRD that may associate to form homodimers, (ii) chimeric, with a single CRD and a large nonlectin amino-terminal domain that contributes to self-aggregation and, (iii) tandem-repeat, with two distinct but homologous CRDs in a single polypeptide [10].

Important to the individual functions of galectins is that each CRD has unique glycan binding preferences that impact their engagement with cell surface receptors and, therefore, signaling function [11]. Here, we studied the ability of three galectins, belonging to each of the three major groups, prototypical, chimeric and tandem-repeat, in modulating CXCL10-mediated chemotaxis.

## 2. Materials and Methods

### 2.1. Cell Culture

Human fibroblasts were derived from five eye bank donor corneas with the approval of the Institutional Review Board of the Massachusetts Eye and Ear Human Studies Committee. Both the epithelium and endothelium were removed by scraping with a razor blade. Primary cultures were prepared by cutting the stromal tissue into small pieces and allowing the explants to adhere to the bottom of tissue culture plates in the presence of Dulbecco’s Modified Eagle Medium /Nutrient Mixture F12 (DMEM/F12; cat. no. 10-092-CV, GE Healthcare) supplemented with 10% newborn calf serum [12]. The fibroblasts were passaged after 1 to 2 weeks cultivation. Telomerase-immortalized human corneal epithelial cells were grown in keratinocyte serum-free medium (cat. no. 17005-042, Thermo Fischer Scientific, Waltham, MA, USA) supplemented with 0.3 mM CaCl_2_, 25 µg/mL bovine pituitary extract, 0.2 ng/mL epidermal growth factor, and 1% penicillin/streptomycin [13]. Human umbilical vein endothelial cells (HUVECs) were cultured in EGM-2 (endothelial growth medium 2; cat. no. CCM027, R&D) supplemented with 20% fetal bovine serum and L-glutamine. THP-1 monocytes and Jurkat T cells were grown in RPMI-1640 (cat. no. 11875135, Gibco^TM^) supplemented with 10% fetal bovine serum. For THP-1 cells, 50 µM of β-mercaptoethanol (cat. no. 21985-023, Thermo Fisher Scientific) was added to the medium.

### 2.2. Expression and Purification of Human Galectins

Expression vectors for human galectin-1, -3 and -8 were transformed into Rosetta^TM^ 2(DE3)pLysS Singles™ competent cells (cat. no. 71401-3, Novagen, Burlington, MA, USA). Heterologous expression of recombinant protein was achieved by treatment with 0.5 m M isopropyl β-D-thiogalactopyranoside for 4 h at 30 °C. An affinity chromatography of β-lactose-conjugated Sepharose column was prepared as previously described to purify the proteins [14]. To eliminate contaminating bacterial endotoxins, the proteins were further purified by polymyxin B affinity chromatography (cat. no. P1411, Sigma-Aldrich, Cambridge, MA, USA). The absence of lipopolysaccharide was confirmed using the ToxinSensor Chromogenic LAL Endotoxin Assay Kit (cat. no. L00350, GenScript, Piscataway, NJ, USA). Protein solutions were concentrated by Amcon^®^ centrifugal filtration 3K (cat. no. UFC900308, Millipore, Burlington, MA, USA), dialyzed against PBS containing 10% glycerol and 5 mM β-mercaptoethanol and stored at −80 °C. The integrity of the recombinant protein was monitored by 10% SDS- polyacrylamide gel electrophoresis.

### 2.3. Hemagglutination Assay

The hemagglutination assay was performed using human red blood cells. Cells were trypsinized and fixed in 1% glutaraldehyde for 1 has described [15]. Fifty μL volumes of 2-fold serially diluted samples were mixed with 10 μL of the erythrocyte suspension, at a final concentration of 1%, in 330-μL round bottom microtiter wells (Corning, Corning, NY, USA). For inhibition assays, the protein was mixed with 0.15 M β-lactose prior to the addition of red blood cells.

### 2.4. Quantitative Real-Time PCR

Total RNA was extracted using the extraction reagent TRIzol (cat. no. 15596026, Thermo Fisher Scientific) and further purified using the RNeasy MinElute Cleanup Kit (cat. no. 74204, Qiagen, Valencia, CA, USA). To remove any residual DNA contamination, RNA was digested using DNase I (cat. no. 89254, RNase-free DNase set; Qiagen). RNA concentration and purity were assessed using a NanoDrop 2000 spectrophotometer (Thermo Fisher Scientific). First strand cDNA was synthesized from 1 μg total RNA using the iScript™ cDNA synthesis kit (cat. no. 1708890, Bio-Rad, Hercules, CA, USA) in a 25-μL reaction volume according to the manufacturer’s instructions. Gene expression was determined by quantitative real-time PCR using the SsoAdvanced™ Universal SYBR^®^ Green Supermix (cat. no. 1725271, Bio-Rad, Hercules, CA, USA) in a Mastercycler RealPlex 2 (Eppendorf, Framingham, MA, USA). Primers for *CXCL10* (cat. no. qHsaCED0034161) and *GAPDH* (cat. no. qHsaCED0038674) were obtained from Bio-Rad. The following parameters were used: 2 min at 95 °C, followed by 40 cycles of 5 s at 95 °C and 30 s at 60 °C. All samples were normalized using *GAPDH* housekeeping gene expression. The comparative 2^−ΔΔCT^ method was used for relative quantitation of the number of transcripts. Non template controls were run in each assay to confirm lack of DNA contamination in the reagents used for amplification.

### 2.5. CXCL10 Secretion Assay

Cells were seeded at a density of 5 × 10^5^ cells/mL in 6-well plates and incubated overnight in their corresponding media. After washing with PBS, cultures were incubated with 50 µg/mL recombinant human galectin (rhGal)-1, -3 and -8 or 50 µg/mL IFN-γ (R&D Systems, Minneapolis, MN, USA) in serum-free DMEM/F12 medium. After 24h, the conditioned media were collected and centrifuged at 13,000× *g* for 10 min at 4 °C to remove cells and cellular debris. CXCL10 concentration was quantified using a Legend Max™ Human CXCL10 ELISA Kit (cat. no. 439904, Biolegend, San Diego, CA, USA) according to the manufacturer’s instructions.

### 2.6. Chemotaxis Assay

Cultures of fibroblasts were incubated with increasing concentrations of recombinant galectins for 24 h. The conditioned media were collected and pre-cleared three times with β-lactose Sepharose beads to remove galectins. The media was placed in the lower compartments of 96-well chambers equipped with 5-μm pore size polyvinylpyrrolidone-free polycarbonate filters (Neuro Probe Inc, Gaithersburg, MD, USA). THP-1 and Jurkat cells were stained for 20 min with 5 µM carboxyfluorescein succinimidyl ester (CFSE; cat. no. 565082, BD Pharmingen, Bedford, MA, USA), then aliquots of 5 × 10^4^ cells in 30 µL of DMEM/F12 containing 1% bovine serum albumin were placed over the assembled filters. The whole chamber was incubated for 2 h protected from light in a humidified incubator with 5% CO_2_ at 37 °C. The chamber was then disassembled and the filters were washed with PBS. Nonmigrating cells on the upper surface of the filters were removed with a cotton swab. Migrating cells on the lower surface of the membrane were detached by incubation with 10 mM EDTA. Total migrating cells from the filter and lower compartment were pooled and centrifuged at 300× *g* for 10 min. The cells were gated according to their forward scatter (FSC) and sideward scatter (SSC) parameters. Then the number of fluorescent cells was measured during a 1 min acquisition at high flow rate using a BD^TM^ LSR II flow cytometer (BD Biosciences, Bedford, MA, USA) as previously described [16]. A mouse monoclonal antibody against CXCL10 (1.2 μg/mL; MAB266, R&D Systems) was added to the lower compartment of the chamber in neutralization studies.

### 2.7. Statistical Analyses

All statistical analyses were performed using Prism 7 (GraphPad Software, San Diego, CA, USA) for Mac OS X.

## 3. Results and Discussion

### 3.1. Galectins Induce CXCL10 in a Cell Type-Specific Manner

Inflammatory disorders are frequently associated with the expression of unique combinations of galectins, known as the ‘galectin signature’, that contribute to shape the local microenvironment [17,18]. Galectin-1, -3 and -8 are members of the galectin family representing the prototypical, chimeric and tandem-repeat types, respectively. Their expression is altered in a number of pathologies that in cornea include infection, transplantation and wound healing [19]. Here, we produced recombinant human protein in *Escherichia coli* to clarify the function of each type of galectin in modulating CXCL10 chemotactic activity. As shown by Coomassie Blue staining, the detected molecular masses of the purified proteins were in agreement with the theoretical molecular masses of each galectin (Figure 1A). Furthermore, each galectin was able to induce hemagglutination of red blood cells in a lactose-dependent manner, supporting functional activity of the recombinant protein and ability to establish carbohydrate-dependent interactions with extracellular glycoconjugates [20].

Most studies evaluating the effect of galectins on cytokine and chemokine production have focused on individual members of the galectin family and their effect on specific cell types. Recent studies have shown that galectin-3 and galectin-8 can promote the expression of CXCL10 in human fibroblasts and mouse osteoblasts, respectively [21,22]. Our experiments indicate that, in addition to these galectins, chimeric galectin-1 also induces expression of CXCL10 in human fibroblasts of corneal origin independently of IFN-γ (Figure 1B). Interestingly, the ability of galectin-1, -3 and -8 to induce CXCL10 secretion was observed primarily in human corneal fibroblasts but not THP-1 monocytes, human corneal epithelial cells or human vascular endothelial cells. We also found differences in the magnitude of the CXCL10 response, with galectin-1 and -8 having a more dramatic effect on CXCL10 expression and synthesis than galectin-3. These results suggest that the actions of galectins on CXC10 levels occur in a tissue-specific manner and include compartments containing matrix-secreting cells. This mechanism of CXCL10 induction on certain cell types would be in contrast with the global actions of IFN-γ, whose receptors can be found on the surface of almost every cell in the body [23]. In future studies it will be interesting to determine the identity of potential galectin counter receptors that initiate CXCL10 signaling in matrix-secreting cells and whether different galectins compete for the same receptors to initiate this response.

### 3.2. Galectins Differentially Dictate Fibroblast-Dependent Immune Cell Chemotaxis

Important to our experiments was determining the biological significance of CXCL10 induction in human corneal fibroblasts. It is now recognized that fibroblasts act as immune sentinels that release chemokines to recruit leukocytes [24,25]. CXCL10 is a potent chemoattractant of CXCR3-expressing cells from the innate and adaptive immune systems [26]. Therefore, in subsequent analyses, we sought to determine how induction of CXCL10 by different galectins would affect the migration of THP-1 and Jurkat cells, two human cell lines extensively used to model monocyte and T cell function, respectively [27,28]. Towards this purpose, we first collected the conditioned media of primary cultures of human corneal fibroblasts treated with increasing concentrations of galectin-1, -3 and -8. Because galectins themselves induce chemotaxis [8], the conditioned media were pre-cleared with β-lactose Sepharose beads to remove galectins prior to the chemotaxis assay (Figure 2A).

Next, we analyzed the potential of the conditioned media to stimulate immune cell chemotaxis. Medium collected from corneal fibroblasts exposed to increasing concentrations of galectin-8 induced a robust migratory response of THP-1 cells but failed to prompt Jurkat cell chemotaxis (Figure 2B). On the other hand, medium collected from fibroblasts exposed to galectin-1 prompted migration of Jurkat cells but not THP-1 cells. The response to galectin-3 was intermediate, slightly affecting the migration of both THP-1 and Jurkat cells. In order to determine whether these migratory activities were influenced by CXCL10, we preincubated the conditioned media with mAb266, a function-blocking antibody against CXCL10. As shown in Figure 2C, addition of mAb266 significantly diminished the number of THP-1 and Jurkat cells that migrated in response to galectin-8 and galectin-1, respectively. By comparison, no significant effects in migration were observed when blocking CXCL10 following addition of galectin-3. These results indicate that the galectin-CXCL10 axis differentially contributes to regulate fibroblast-dependent immune cell chemotaxis.

A major question raised by the present findings is how galectins contribute to the differential migration of THP-1 and Jurkat cells after induction of CXCL10. There are several potential explanations for our observation. Other than CXCL10, fibroblasts produce factors that condition the cellular and cytokine microenvironment [29]. Among them are matrix metalloproteinases that modify cytokines and chemokines and consequently affect leukocyte migration [30]. For instance, cell culture supernatants obtained from human cardiac fibroblasts contain MMP2 that degrades the CCL7 chemokine and impairs the chemotaxis of THP-1 monocytes [31]. Also, fibroblasts can express stimulatory signals that directly regulate the activation and differentiation of T cells and, consequently, their trafficking [32,33]. It is likely that members of the galectin family, by differentially affecting the expression of these other factors, intricately regulate the CXCL10-dependent migration of immune cells. Further experiments based on high throughput biological systems will be needed to unravel this complex regulatory network.

## 4. Conclusions

We find that, independently of their classification, galectins influence CXCL10 synthesis mostly in human fibroblasts. We continue to show that each galectin appears to have a differential effect in the CXCL10-mediated mobilization of monocytes and T cells. These findings highlight the potential importance of the galectin signature in enhancing the paracrine action of fibroblasts and dictating the recruitment of specific subsets of immune cells into stromal compartments.

## Figures and Tables

**Figure 1 cells-10-00274-f001:**
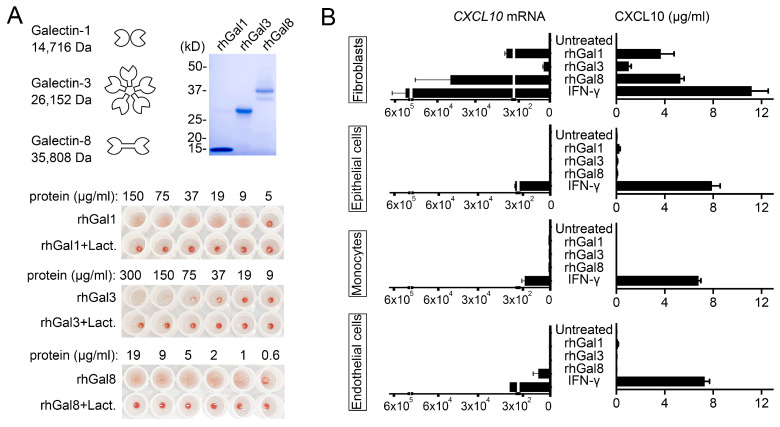
Galectins induce CXCL10 in a cell type-specific manner. (**A**) SDS-polyacrylamide gel electrophoresis of recombinant human galectins (rhGal) stained with Coomassie Blue. Molecular weights of protein standards are indicated on the left. The hemagglutination assay was performed using serial two-fold dilutions of the recombinant protein. The bottom row contains protein incubated with β-lactose, a competitive carbohydrate inhibitor of galectin binding. (**B**) Cultures of human corneal fibroblasts, human corneal epithelial cells, THP-1 monocytes and human umbilical vein endothelial cells (HUVECs) were incubated with 50 µg/mL rhGal-1, -3 and -8 or 50 µg/mL IFN-γ. Quantitative real-time PCR was used to determine the levels of *CXCL10* mRNA after 6 h of incubation (*n* = 3 independent experiments), whereas ELISA was used to determine the levels of CXCL10 protein in cell culture supernatants after 24 h of incubation (*n* = 6 independent experiments). The data represent the mean ± SEM.

**Figure 2 cells-10-00274-f002:**
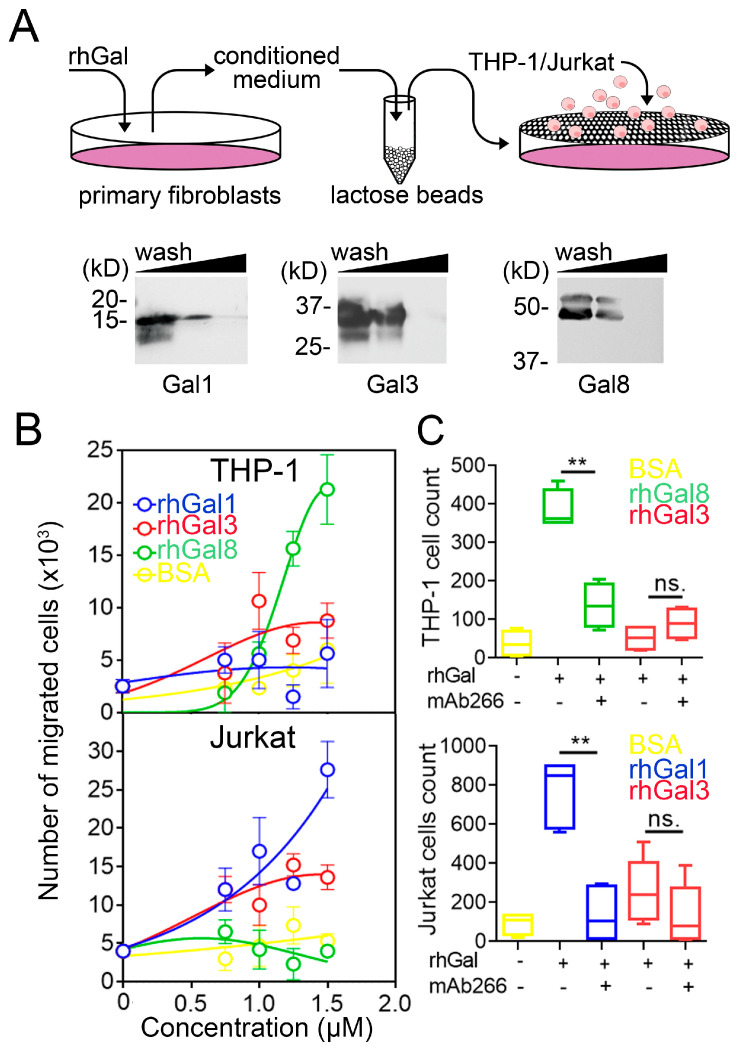
Galectins differentially dictate fibroblast-dependent immune cell chemotaxis. (**A**) Recombinant human galectin (rhGal)-1, -3 and -8 were incubated with fibroblasts for 24 h. The conditioned media were pre-cleared with β-lactose Sepharose beads three times to remove galectins prior to chemotaxis assays. The presence of galectins in pre-cleared conditioned media was evaluated by immunoblotting. (**B**) Quantification by flow cytometry of the migration of CFSE-labeled THP-1 and Jurkat cells to conditioned media from fibroblasts treated for 24 h with increasing concentrations of recombinant galectins or BSA (*n* = 3 independent experiments performed in duplicate). (**C**) Migration of CFSE-labeled THP-1 and Jurkat cells to conditioned media from fibroblasts treated for 24 h with 50 μg/mL of recombinant galectins or BSA. A CXCL10 neutralizing antibody was added to the conditioned media in neutralization studies (*n* = 4–5 independent experiments in duplicate). The data in (**B**) represent the mean ± SEM. The box and whisker plots show the 25 and 75 percentiles (box), the median, and the minimum and maximum data values (whiskers). Significance was determined using Student’s t test. ** *p* < 0.01.

## Data Availability

The data presented in this study are available in article.

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
