# Peer review of "Induction of CXCL10-Mediated Cell Migration by Different Types of Galectins"

_cells, 2021, doi:10.3390/cells10020274_

Round 1

Reviewer 1 Report

In this report, “Induction of CXCL10-Mediated Cell Migration by Different Type of Galectins”, the authors examined the effects of galectins -1, -3, and -8 on inducing chemokine interferon γ inducible protein 10 (CXCL10) independently of interferon γ, while mediating migration in various cell lines. Results indicated that CXCL10-induced galectins was found to be prominent on stromal fibroblast isolates from human cornea and had no effect on epithelial, monocytic, and endothelial cells. Moreover, induction of CXCL10 by galectin-8 showed increased cell migration response on THP-1 monocytes, while galectin-1 mediates Jurkat cells migration.

Minor points:

  • Line #26 The word eighties should be switch to nineteen eighties
  • Isotype staining for mAb266
  • CAT # should be added for product and/or reagent
  • There are spelling errors and typos in the paper. e.g. “No template” in (#104) ® Non template, Figure 1 is labeled as 2 in the figure legend.
  •  

Major points:

  • Figure 2B, there is no CFSE assay on untreated THP-1 and Jurkat cells, expression controls are needed for correct interpretation of CFSE dye peak. How can it be established that the THP-1 and Jurkat cell migration was due to the CXCL10 released in the condition media when galectins could induce other molecules and/ or fibroblast release different factors that can condition cell migration?
  • No explanation on Flow cytometry protocol on method section.
  • A gating strategy showing live dead for CFSE assay is needed as the dye's high concentration is known to kill cells.

Author Response

Reviewer 1

In this report, “Induction of CXCL10-Mediated Cell Migration by Different Type of Galectins”, the authors examined the effects of galectins -1, -3, and -8 on inducing chemokine interferon γ inducible protein 10 (CXCL10) independently of interferon γ, while mediating migration in various cell lines. Results indicated that CXCL10-induced galectins was found to be prominent on stromal fibroblast isolates from human cornea and had no effect on epithelial, monocytic, and endothelial cells. Moreover, induction of CXCL10 by galectin-8 showed increased cell migration response on THP-1 monocytes, while galectin-1 mediates Jurkat cells migration.

Minor points:

Line #26 The word eighties should be switch to nineteen eighties. We have modified the manuscript according to the reviewer’s suggestion.

Isotype staining for mAb266. We have performed staining of the isotype control for mAb266 (mAb266) in both Jurkat and THP-1 and we have not noticed any nonspecific interaction (data not shown). We are aware that using the isotype control antibody is ideal as negative control for CXCL10 neutralization assay, but instead we used fibroblast conditioned medium without the blocking antibody as a negative control. Use of this negative control is common and considered acceptable for this kind of assay [1,2].

CAT # should be added for product and/or reagent. We have now included the catalog number for the reagents and products used in the Materials and Methods section.

There are spelling errors and typos in the paper. e.g. “No template” in (#104) ® Non template, Figure 1 is labeled as 2 in the figure legend. We apologize for this oversight. The errors have been corrected as advised.

Major points:

Figure 2B, there is no CFSE assay on untreated THP-1 and Jurkat cells, expression controls are needed for correct interpretation of CFSE dye peak. Certainly, we performed this control but we did not include it originally for clarity purposes. As suggested by the reviewer, we have modified Fig. 2B to include the chemotaxis data of THP-1 and Jurkat toward BSA-treated fibroblasts conditioned medium.

We would like to note that CFSE staining was used to discriminate migrated cells from debris for counting purposes using flow cytometry. One hundred % of THP-1 and Jurkat cells were stained with CFSE before being subjected to migration toward treated fibroblast conditioned medium. Counting of CFSE stained cells by flow cytometry was performed according to their forward scatter (FSC) and sideward scatter (SSC) parameters during a 1 min acquisition at high flow rate as previously described [3]. To further validate our counting technique, we counted the cells under the microscope using Trypan blue exclusion stain.

How can it be established that the THP-1 and Jurkat cell migration was due to the CXCL10 released in the condition media when galectins could induce other molecules and/ or fibroblast release different factors that can condition cell migration? This concern was addressed by using a blocking antibody (mAb266) to neutralize CXCL10 activity in conditioned media from treated fibroblasts. We now include additional data indicating that blocking CXCL10 influences the chemotactic effects of galectin-1 and -8 but not -3 (Fig. 2C).

No explanation on Flow cytometry protocol on method section. We now better explain the counting protocol used in flow cytometry experiments.

A gating strategy showing live dead for CFSE assay is needed as the dye's high concentration is known to kill cells. We thank the reviewer for raising this concern. It was indeed raised during the initial planning of our experiments. For that, we performed experiments to optimize the concentration of CFSE that showed no effect on cell viability. Here, we used Trypan blue exclusion to distinguish between live and dead cells. The concentration of 5 μM CFSE was non-toxic for THP-1 and Jurkat cells, consistently with previous reports [4-6]. Furthermore, the cells were gated according to their forward scatter (FSC) and sideward scatter (SSC) parameters, then counted for CFSE positive cells during a 1 min acquisition at high flow rate. To further validate the FACS counting, a manual one was performed under the microscope using Trypan blue. Both procedures of counting were comparable.

References:

  1. Dominguez, F.; Martinez, S.; Quinonero, A.; Loro, F.; Horcajadas, J.A.; Pellicer, A.; Simon, C. CXCL10 and IL-6 induce chemotaxis in human trophoblast cell lines. Mol Hum Reprod 2008, 14, 423-430, doi:10.1093/molehr/gan032.
  2. Agostini, C.; Calabrese, F.; Rea, F.; Facco, M.; Tosoni, A.; Loy, M.; Binotto, G.; Valente, M.; Trentin, L.; Semenzato, G. Cxcr3 and its ligand CXCL10 are expressed by inflammatory cells infiltrating lung allografts and mediate chemotaxis of T cells at sites of rejection. Am J Pathol 2001, 158, 1703-1711, doi:10.1016/S0002-9440(10)64126-0.
  3. Baj-Krzyworzeka, M.; Majka, M.; Pratico, D.; Ratajczak, J.; Vilaire, G.; Kijowski, J.; Reca, R.; Janowska-Wieczorek, A.; Ratajczak, M.Z. Platelet-derived microparticles stimulate proliferation, survival, adhesion, and chemotaxis of hematopoietic cells. Exp Hematol 2002, 30, 450-459.
  4. Jin, Q.; Jiang, L.; Chen, Q.; Li, X.; Xu, Y.; Sun, X.; Zhao, Z.; Wei, L. Rapid flow cytometry-based assay for the evaluation of gammadelta T cell-mediated cytotoxicity. Mol Med Rep 2018, 17, 3555-3562, doi:10.3892/mmr.2017.8281.
  5. Czernek, L.; Chworos, A.; Duechler, M. The Uptake of Extracellular Vesicles is Affected by the Differentiation Status of Myeloid Cells. Scand J Immunol 2015, 82, 506-514, doi:10.1111/sji.12371.
  6. Yan, Y.; Chang, L.; Tian, H.; Wang, L.; Zhang, Y.; Yang, T.; Li, G.; Hu, W.; Shah, K.; Chen, G., et al. 1-Pyrroline-5-carboxylate released by prostate Cancer cell inhibit T cell proliferation and function by targeting SHP1/cytochrome c oxidoreductase/ROS Axis. J Immunother Cancer 2018, 6, 148, doi:10.1186/s40425-018-0466-z.

Reviewer 2 Report

In this paper, entitled “Induction of CXCL10-Mediated Cell Migration by Different Types of Galectins”, AbuSamra et al. describe the role of various galectins in induction of CXCL10, which they link to promotion of cell migration in THP and Jurkat model cell lines. This work adds to an extensive existing body of work focused on the role of various members of the galectin family in immune regulation. The study is well designed and addresses an interesting question.

Neutralization studies with Gal-3 would provide a nice complement to the neutralization studies with Gal-1 and Gal-8 treatment. Also, while the effects of conditioned sera from Gal-1 treated fibroblasts seem to be completely neutralized, neutralization of media from Gal-8 conditioned fibroblasts is less complete. Do the authors feel the residual activity is due to limitations of the assay or non-CXCL10 related activity. Is blocking dose dependent and does the blocking activity shown represent the maximum blocking achieved regardless of increased doses of blocking antibody?

Did the authors assess the differential effects of conditioned media from HUVECs treated with various galectins? This may be an interesting comparison.

Minor comments:

The figure legend of Figure 1 is mislabeled as figure 2.

Author Response

Reviewer 2

‘’In this paper, entitled “Induction of CXCL10-Mediated Cell Migration by Different Types of Galectins”, AbuSamra et al. describe the role of various galectins in induction of CXCL10, which they link to promotion of cell migration in THP and Jurkat model cell lines. This work adds to
an extensive existing body of work focused on the role of various members of the galectin family in immune regulation. The study is well designed and addresses an interesting question.
We thank reviewer 2 for his constructive comments and overall favorable assessment of our manuscript.

Neutralization studies with Gal-3 would provide a nice complement to the
neutralization studies with Gal-1 and Gal-8 treatment.
We have modified Fig. 2C to include the neutralization results obtained with galectin-3 as suggested.

Also, while the effects of conditioned sera from Gal-1 treated fibroblasts seem to be completely neutralized, neutralization of media from Gal-8 conditioned fibroblasts is less complete. Do the authors feel the residual activity is due to limitations of the assay or non-CXCL10 related activity. Is blocking dose dependent and does the blocking activity shown represent the maximum blocking achieved regardless of increased doses of blocking antibody? We certainly acknowledge that other chemokines could contribute to migration independently of CXCL10. In our experiments we optimized the concentration of the mAb266 blocking antibody to reach the maximum degree of CXCL10 neutralization. We apologize from the confusion due to using different graph parameters, but the number of migrated cells after CXCL10 neutralization were similar in both THP-1 toward rhGal-8 treated sera and Jurkat toward rhGal-1 treated sera with (avg. 136 and 141 respectively). 

Did the authors assess the differential effects of conditioned media
from HUVECs treated with various galectins? This may be an interesting
comparison.
We certainly agree that this could be an interesting comparison. In our communication, we focused on fibroblasts as they show the highest magnitude of CXCL10 response toward galectin treatment.

Minor Comment
The figure legend of Figure 1 is mislabeled as figure 2.’’ We apologize for this oversight. This has been corrected.

Reviewer 3 Report

The aim of this communication is to clarify the function of galectins in modulating CXCL10 using Gal-1, -3 and -8 to represent each of the 3 major classifications of galectins. The first contribution of this paper CXCL10 induction by Gal1 and Gal8 was cell type dependent, specifically in endothelial cells. Secondly, Gal1 or Gal8 induction of CXCL10 resulted in chemotactic migration of different immune cells (monocytes / T-cells).

The experiments and design are sound. The implications in cancer research and diseases that involve chronic inflammation are novel due to connections involving epithelial to mesenchymal transition and immune response. It is novel that the induction of CXCL10 is tissue and immune cell specific which implies sub-compartmental ECM fibrotic deposition. The connection you make in the introduction to diseases needs to revised in the conclusions. But overall well written and novel.

I only found a couple of editing errors. Line 15 “be-came in contact” should be came in contact. Figure 1 is mislabeled as figure 2.

Lines 50-52: sentence needs to be slightly reworded for clarity. From “Here, we studied the ability of three galectins, belonging to the prototypical, chimeric and tandem-repeat groups, in….” to Here, we studied the ability of three galectins, belonging to the each of the three major groups, prototypical, chimeric and tandem-repeat groups, in….” or something similar.

Figure 1A blot, there seems to be some contaminants in the Coomassie especially in Gal3 and 8.

Figure 1B the 2x102 gap should be adjusted so the mRNA levels for IFN-gamma can be seen for the epithelial, monocytes and endothelial cells.

Is research design appropriate. I would like to see a set of experiments with conditioned media from endothelial or epithelia cells for Figure 2B and C or just C as a negative control.

Author Response

Reviewer 3

The aim of this communication is to clarify the function of galectins in modulating CXCL10 using Gal-1, -3 and -8 to represent each of the 3 major classifications of galectins. The first contribution of this paper CXCL10 induction by Gal1 and Gal8 was cell type dependent, specifically in endothelial cells. Secondly, Gal1 or Gal8 induction of CXCL10 resulted in chemotactic migration of different immune cells (monocytes / T-cells).

The experiments and design are sound. The implications in cancer research and diseases that involve chronic inflammation are novel due to connections involving epithelial to mesenchymal transition and immune response. It is novel that the induction of CXCL10 is tissue and immune cell specific which implies sub-compartmental ECM fibrotic deposition. The connection you make in the introduction to diseases needs to revise in the conclusions. But overall well written and novel. We thank reviewer 3 for his constructive comments and overall favorable assessment of our manuscript.

I only found a couple of editing errors. Line 15 “be-came in contact” should be came in contact. Figure 1 is mislabeled as figure 2. We apologize for this oversight. These errors have been corrected.

Lines 50-52: sentence needs to be slightly reworded for clarity. From “Here, we studied the ability of three galectins, belonging to the prototypical, chimeric and tandem-repeat groups, in….” to Here, we studied the ability of three galectins, belonging to the each of the three major groups, prototypical, chimeric and tandem-repeat groups, in….” or something similar. We have modified the manuscript to be more comprehensive.

Figure 1A blot, there seems to be some contaminants in the Coomassie especially in Gal3 and 8. In our preparation procedure, we assure that the purity of produced recombinant human galectin-1 and -3 are over 95% by SDS-PAGE Coomassie blue staining. Producing recombinant human galectin-8 is more difficult, and our values vary between 75 and 80%. One reason that explains the reduction in protein purity is the high tendency to precipitation of galectin-8, which forces us to use harsher conditions in the procedure that could lead to degradation. Indeed, commercially available rhGal-8 from Elabscience, SinoBiological and Cusabio show similar molecular weight contaminant bands by SDS-PAGE Coomassie blue staining. Also, it is possible to obtain Millard reaction products, since the sample contains rhGal-8 and sugars that are boiled in reducing buffer sample.

Figure 1B the 2x102 gap should be adjusted so the mRNA levels for IFN-gamma can be seen for the epithelial, monocytes and endothelial cells. We have fixed the graph as suggested.

Is research design appropriate. I would like to see a set of experiments with conditioned media from endothelial or epithelia cells for Figure 2B and C or just C as a negative control. We focused on fibroblasts as they show the highest magnitude of CXCL10 response toward galectin treatment. It is true that galectins exert chemotactic effects through other mediators. Our data go along with previous research that highlighted the role of Gal-8 in inducing the production of various chemokines and cytokines (CCL2, CXCL3, CXCL8, CXCL1, GM-CSF, IL-6 and CCL5) using cytokine arrays. The research revealed that these molecules especially CCL5 are involved in recruiting inflammatory cells like platelets. Interestingly CXCL10 was not recognized to play a role in such processes as it was not detected in the cytokine array [1]. 

References:

  1. Cattaneo, V.; Tribulatti, M.V.; Carabelli, J.; Carestia, A.; Schattner, M.; Campetella, O. Galectin-8 elicits pro-inflammatory activities in the endothelium. Glycobiology 2014, 24, 966-973, doi:10.1093/glycob/cwu060.

Round 2

Reviewer 1 Report

Reviewers adequately addressed recommendations

Reviewer 3 Report

Thank you for addressing the issues noted in the review of your manuscript. I believe the manuscript is publishable in the current form.